# Physical Activity and Anxiety of Chinese University Students: Mediation of Self-System

**DOI:** 10.3390/ijerph18094468

**Published:** 2021-04-22

**Authors:** Sumaira Kayani, Tayyaba Kiyani, Saima Kayani, Tony Morris, Michele Biasutti, Jin Wang

**Affiliations:** 1Department of Psychology, College of Education, Zhejiang Normal University, Jinhua 321001, China; sumaira@zjnu.edu.cn; 2Department of Education, Pir Mehr Ali Shah Arid Agriculture University Rawalpindi, Punjab 46300, Pakistan; 3Department of Physical Education, Zhejiang University, 866 Yuhangtang Road, Hangzhou 310058, China; kiyani@zju.edu.cn; 4Department of Education, Women University Azad Jammu and Kashmir, Bagh 12500, Pakistan; saimakayani22@gmail.com; 5Institute of Health and Sport, Victoria University, Ballarat Rd, Footscray, VIC 3011, Australia; tony.morris@vu.edu.au; 6Department of Philosophy, Sociology, Education and Applied Psychology (FISPPA), University of Padova, 35139 Padova, Italy

**Keywords:** physical activity, students’ anxiety, self-concepts, the mediating role

## Abstract

The present study examined the role of self-enhancement and self-criticism in the relationship between physical activity and anxiety. A total of 305 students from Chinese universities, ranging in age from 18 to 36, completed a questionnaire package comprising a physical activity questionnaire, a self-enhancement strategies scale, a level of self-criticism scale, and a short form of state and trait anxiety scale. Findings highlighted that physical activity had a significant negative correlation with anxiety (r = −0.31, *p* < 0.01), a significant positive association with self-enhancement (r = 0.43, *p* < 0.01), and a significant negative relationship with self-criticism (r = −0.14, *p* < 0.05). It was also found that anxiety was significantly predicted by self-enhancement (−0.21, *p* < 0.01) and self-criticism (0.44, *p* < 0.01). Moreover, the mediation model supports the mediation of self-enhancement and self-criticism between physical activity and anxiety in university students. The findings suggest that interventions aimed at promoting physical activity and enhancing the self-system should be worthy strategies for reducing students’ anxiety.

## 1. Theoretical Background

In China, scholars are examining ways to help people cope with the stresses of modern life that affect their mental health. Anxiety is a common mental issue characterized by a complex emotional reaction to stressful circumstances affecting psychological and behavioral states [1,2]. Various studies conducted in China reveal that physical activity (PA) has a positive relationship with psychological well-being [3,4,5]. PA means body movement in which skeletal muscles are involved [6], and there are evidences that PA is beneficial for decreasing anxiety and pressure that often affects students’ cognitive performance (Biddleand Asare [7]. In various empirical studies involving Chinese students, researchers have found PA to be positively associated with a reduction in anxiety, stress, and depression [8,9]. Nevertheless, the rate of participation in PA and sport is declining nowadays in China [10], as 83.8% of those aged 18 or older are inactive in China [11].

The relationship between PA and anxiety has been an important topic considered by educational psychologists, sports psychologists, and medical experts for many years [12]. Many researchers have found that individuals performing more PA have greater emotional stability [13], are less prone to feel depression [14], are less likely to experience anxiety [8], feel greater self-esteem, and have a lower risk of mental illness [15,16]. Most of the research has paid attention to the association of PA with mental health, psychological well-being and self-concept, with different samples from China [3,17,18], but a few studies have mentioned the connection between PA and the anxiety of university students [15]. Hence, the relationship between PA and the anxiety of university students still needs to be investigated in greater detail [8,19,20].

Theoretically, several theories and models address the relationship between PA and psychological well-being [21]. For instance, researchers who developed the biopsychosocial model (BPSM) of physical and mental health proposed that PA promotes mental health and reduces depression and anxiety [22,23]. Social withdrawal theory [24] explains that physical inactivity is directly associated with psychosocial problems, including stress, depression, or anxiety. Research empirically supports the claim that anxiety is increased when PA is scarce or not performed at all by students [25]. Conversely, Lee et al. reported that physically active students had lower anxiety than students who were less physically active [26].

Several studies have examined the effect of PA and exercise on depression and anxiety among university students [15,27]. Li, Xu [15] pointed out that PA was a protective factor for the academic stress of students, and Wang, Li [27] confirmed, through an experimental study, that long-term PA significantly prevents anxiety and depression among university students. Despite numerous studies linking PA and reduced anxiety, a recent review in China suggested investigating the mechanism/s underlying specific aspects, such as the intermediary role of other variables, including self-concept [8]. This raises the proposition that there are other pathways from PA to anxiety. For instance, PA may indirectly affect anxiety through other factors.

The influence of PA on anxiety is likely to be mediated by different social, psychological, cognitive, and psychosocial factors. However, the mechanism through which PA transfers its effect to anxiety is poorly understood. One potential mediator between PA and anxiety is self-enhancement (SE), which is a positive personality trait or a cognitive propensity [28] for maximizing positive self-views and minimizing negative self-views [29]. SE is considered to be an essential element for mental well-being and psychological prosperity [30]. Researchers have found that SE is positively associated with PA and inversely linked with mental illness [18,31]. Previous studies have mentioned SE as a protective therapy that promotes anxiety reduction [32,33,34] and psychological well-being [35]. In a recent meta-analysis, Dufner, Gebauer [36] concluded that participants who have positive emotions and happy moods are less likely to feel anxious. In addition, physically active individuals usually feel self-enhanced, as compared to less active and depressed populations [37]. Health, productive lifestyle, well-being, positive feelings, and enhancement can be maintained and improved through PA [25,38,39]. PA develops positivity and emotional stability among young people and reduces barriers through which well-being can be enhanced [40]. From the literature reported here, it is inferred that SE could be a mediator of the relationship between physical health and psychological well-being [41]. However, the mediating role of SE between PA and anxiety has not yet been examined in a Chinese context.

Another potential mediator between PA and anxiety is self-criticism (SC). SC is a negative personality trait and is characterized by possessing negative views of the self [42]. Duarte, Stubbs [43] reported that SC has been found to be associated with anxiety disorders in the literature. SC is conceptualized as negatively affecting psychological and mental health, specifically depression, anxiety, and affective disorders [44,45]. People with high SC do not feel satisfied with themselves, nor do they appreciate themselves [46,47] Their negative thinking and anxiety are increased in various situations [48]. For example, self-critical individuals develop unhealthy perfectionism that leads to psychological distress [49]. Furthermore, students possessing SC are found to be inactive or less active, leading them to have more anxiety [50]. Therefore, when individuals exhibit SC, it should be decreased through contemporary approaches and possible solutions, as little is known about how to reduce SC [46]. Epidemiologic studies have posited that PA can protect people from depression, and individuals who engage in higher levels of PA tend to display lower SC [51], as SC has previously been shown to be negatively related to PA (Kayani et al., 2018). Adverse effects of SC, depression, and anxiety can be reduced through continued PA [52,53]. This theoretical explanation demonstrates that SC is another variable that has the potential to explain the effect of PA on anxiety.

Further, a study on Chinese adolescents provides evidence that PA can be associated with a positive change in moods [54], which reduces stress and anxiety levels [55]. PA and exercise have been proposed to be useful for the promotion of aspects of self-concept, including self-efficacy and the enhancement of moods [8], which could be beneficial for mental health. From a psychological perspective, it is understandable that increasing physical health enhances mood, which reduces stress and anxiety, thus improving mental health. While a positive self-concept has been shown to promote well-being, a negative self-concept causes depression and anxiety. 

We are not aware of any study that provides empirical evidence and elucidates the theoretical relationships among PA, SE, SC, and anxiety in a concurrent model in the Chinese context. We have found only two exceptions in the extant literature addressing PA, self-system, and anxiety issues in a concurrent model. However, these studies were conducted in Canada [50] and Pakistan [56], where psychological well-being, and self-concepts are perceived as different from those in China. Research has consistently reported lower well-being, lower self-concept and life satisfaction, and greater depression and anxiety by Easterners such as the Chinese, Japanese, and Koreans, compared to Westerners such as the British, Americans, and Canadians (Cai, Wu, Shi, Gu, and Sedikides, 2016; Hepper et al., 2013). For example, Cai, Brown, Deng, and Oakes (2007), in a sample of university undergraduates, found that Chinese participants possessed fewer positive cognitive self-evaluations than Americans. Likewise, a few recent studies have explored how Westerners are more self-enhancing while eatserners are more self-critical (Cai, Wu, Shi, Gu, and Sedikides, 2016; Hepper et al., 2013). Besides, Canada and Pakistan follow the independent and interdependent self [57,58]. On the other hand, according to the bi-cultural self-theory of contemporary Chinese individuals, the Chinese self-system possesses both individual and social self orientations [59]. Additionally, the biopsychosocial model argues that the self-concept is a protective cognitive strategy for anxiety issues, and also proves to be a supportive mediating factor between physical exercise and mental well-being (Di Benedetto, 2010, 2015). However, the authors studied other self-concepts such as self-esteem and self-efficacy, but not self-enhancement and self-criticism (Di Benedetto, 2015). This has led us to the expectation that these self-systems would play a different role between physical activity and students’ anxiety in China. Therefore, in this study, we have investigated the role of SE and SC as mediators between PA and students’ anxiety. We have predicted that the self-system, particularly SE and SC, would interevene between PA and students’ anxiety.

### Current Study

In the present study, we investigated the dynamic association between PA, anxiety, SE, and SC. We developed a hypothetical model by integrating the BPS model components describing PA as a beneficial coping strategy for reducing depressive symptoms and anxiety issues [22,23]. We proposed that the self-system (SE and SC) plays a role in the biopsychosocial relationship between PA and anxiety. We proposed that SE and SC would predict the anxiety of students. We hypothesized that students doing more PA would become less anxious. We further predicted that students doing PA would become more self-enhanced and less self-critical. We then predicted that possessing a positive self-concept and avoiding negative self-views would lead to less anxiety. Specifically, we hypothesized that PA and anxiety would be mediated through SE and SC. 

To control for additional variables affecting PA, SE, SC, and anxiety of students, we included age, education, and gender as covariates. Past studies have consistently reported that male students are more active than female students [60,61]. Further, several studies have concluded that PA and self-concept are higher in males than in females [62]. In addition, we can argue that students studying in higher degrees would find less time to undertake exercise and PA than those who study in lower degrees, yet less education is described as directly associated with physical inactivity in the literature [63]. Moreover, younger students are found to be more active than older ones in the past research [61]. The study model is presented in Figure 1.

## 2. Method

### 2.1. Measurement of Variables 

#### 2.1.1. Physical Activity

PA was measured through various tools, including objective and subjective measures. Subjective measures included logbooks, self-reports, questionnaires, and diaries of activity, while the objective tools included pedometers, accelerometers, and heart rate monitoring and oxygen consumption [64]. These measures were readily applicable to a large population, and all of them did not fulfill the requirements of the recommendations of type, intensity, frequency, duration, and length of PA. The duration, frequency, and quantification of the intensity of PA performed were required for the present study. Therefore, Cho’s five items PA questionnaire (PAQ) was specially developed on the basic principles of frequency, duration time, intensity, overall length, and type of PA for the current research [65]. This is a valid and reliable measure based on previous studies [66]. It had five items about the type, frequency, intensity, duration, and total length of PA, measured on a 5-point Likert scale. Example questions were, “during a week, how often do you participate in the activity in your free time?”; and “how intensely do you participate in the activity?” The tool was validated in an East Asian country before, but it was revalidated in the Chinese context. To determine the PA level, we first obtained a score by summing the responses on overall length, frequency, duration, and intensity of PA. The summed scores were then multiplied by the scaled score on the type of PA performed by the students. The minimum score was 4 and the maximum score was 100. The levels of PA were categorized into a very high level greater than 96; a high level between 64 and 95; an acceptable level between 36 and 63; a low activity level between 16 and 35; and an inactive level between 4 and 15. Higher scores indicated higher activity levels, while lower scores indicated low activity levels. 

#### 2.1.2. Self-Enhancement

We measured SE by using the short-form of SE Strategies scale (SFSESS; Hepper et al., 2013), with the permission of the corresponding author. It was a validated 20-item scale rated on a 5-point Likert scale (1 = not at all characteristic of me, 5 = very characteristic of me). A sample item was: “When you achieve success or really good grades, do you think it was due to your ability?” Further, internal consistency reliability for the current data was α = 0.95. 

#### 2.1.3. Self-Criticism

The Level of SC scale (LOSC) was adopted for the study. The LOSC is a 22-item scale developed by Thompson and Zuroff [67] to measure two aspects of negative SC in undergraduate students. Twelve items measured one dimension named comparative SC (CSC). A sample item was “I have a nagging sense of inferiority”. A further 10 items measured another dimension named internalized SC (ISC). A sample item was “Failure is a very painful experience for me.” The items 6, 8, 11, 12, 16, 20, and 21, made positive statements, so they were reverse coded. Items were rated on a 5-point Likert scale, on which 1 = not at all, and 5 = very well. Thus, the total score range was 22 (1 × 22 items) to 110 (5 × 22 items). We translated the LOSC into Chinese, using the translation and back-translation method [68] and it was revalidated with a Chinese student sample, as no Chinese version was available. 

#### 2.1.4. Anxiety

To test anxiety we used the short form of the state scale of Charles Spielberger’s State-Trait Anxiety Inventory (STAI Y-6)[69], which is a widely used, valid, and reliable measure for assessing anxiety in various cultures [70,71,72]. This short form of state scale contained six items. The items were, “I am tensed.”, “I feel calm”, and I am relaxed”, “I feel content”, “I feel upset”, and “I am worried”. The scale was measured on a 5-point Likert scale from 1 = not at all to 5 = very much. The calm, relaxed, and content items were reverse coded. 

### 2.2. Data Analysis

Using SPSS version 20, data were checked for normality. Then, means and standard deviations were calculated to present a basic description of the study variables. After that, correlations among the variables were calculated to provide a basis for testing the model by indicating whether the variables that were proposed to be related in the model do actually correlate with each other. The measurement model was developed in AMOS graphics version 23. Model fit indices were estimated in terms of χ2/df ≤ 5, SRMR ≤ 0.08, RMSEA ≤ 0.08, CFI ≥ 0.90, TLI ≥ 0.90 [73,74]. Major hypotheses were tested in Hayes’ PROCESS ver. 3 (2018) by developing multiple mediation models and performing bootstraps. The advantageous characteristic of multiple mediation models was that they allowed us to determine “to what extent specific M (mediator) variables mediate the X/Y (X = independent variable; Y = dependent variable) effect, conditional on the presence of other mediators in the model” [75].

## 3. Results

### 3.1. Demographic Information

Participants were Chinese university students who were accessed by adopting a convenience sampling technique. We distributed a set of questionnaires among 700 students during the fall semester of 2018. We received 504 questionnaires back within two months. The response rate was 72%. A total of 148 questionnaires were useless in the received pack, as they exhibited random responses. It also includes the cases with missing values that were minimal, hence, excluded from the data. Mahalanobis distance was calculated for identifying outliers, and 51 cases were excluded from the analysis on the suspicion of being distorted. Finally, 305 (age range = 18–36) cases were included in the analyses of the study. A total of 185 (60.7%) male students and 120 (39.3%) female students participated. In total, 86 (28.2%) students were from a bachelor’s program, 139 (45.6%) were from a master’s program, and 80 (26.2%) were from a Ph.D. program. Students from ages 18 to 25 made up 32.5% (*n* = 99), 26 to 30 years comprised 37% (*n* = 113), 31 to 35 made up 25.9% (*n* = 79), and 36 and above comprised 4.6% (*n* = 14) of the total population. Most of the students were from social sciences (*n* = 113, 37%) and computer and engineering fields (*n* = 105, 34.4%), and others were from natural and applied sciences (*n* = 87, 28.5%).

### 3.2. Preliminary Analyses 

Basic analyses indicate that data for all variables are normally distributed, with skewness ranging from −0.16 to −1.26, and Kurtosis ranging from −0.16 to 1.44 [76]. Further analyses were performed in three steps. Exploratory factor analysis (EFA) was performed by randomly taking half of the sample. It exhibited four individual factors for PA, SE, and SC and anxiety, with a total variance of 64.42% and with an eigenvalue above one. The first factor explained 27.92% variance, which confirmed that our data set was not threatened by common method bias, as the first factor did not explain a major percentage of the variance [77]. Common method bias was also checked by comparing the standardized regression weights of the model with common latent factor (CLF) and without CLF [78]. There was no significant difference between the two, indicating no path was affected by common method bias. 

### 3.3. Correlation among Study Variables

Table 1 shows mean and standard deviation values and bivariate correlations among all variables. In this sample, PA was negatively associated with students’ anxiety, SE was positively related to PA, while SC was negatively linked with PA. This means that students with favorable views of the self, performed more PA than those with less favorable views. Conversely, students possessing more negative views of the self were less active or inactive compared to students who were less self-critical.

### 3.4. Validity and Reliability

Loading range for the PAQ was 0.729 to 0.840; 0.70 to 0.88 for the SFSESS; 0.70 to 0.90 for the LOSC, and 0.73 to 0.91 for STAI Y-6. Confirmatory factor analysis (CFA) was then applied to assess the standardized factor loadings, validity, and reliability of the factors, measurement, and structural model fit. PAQ, SFSESS, LOSC, and STAI Y-6 were valid and reliable before use. All the items showed satisfactory standardized factor loadings (above 0.70, *p* < 0.001). Composite reliability (CR) was above 0.70. Convergent validity was measured by calculating the average variance extracted (AVE). Convergent validity (AVE > 0.50, and <CR) and discriminant validity (AVE > MSV; √AVE > the inter-construct correlations) provided satisfactory results [73]. The model fit was acceptable, as we found the data fit the measurement model (χ2/df = 2.13, CFI = 0.92, TLI = 0.91 RMSEA = 0.06, and SRMR = 0.05) [73,74].

### 3.5. Results of the Tested Pathways

To test the major hypotheses and to examine the direct and indirect relationships between PA and anxiety through SC and SE, we developed multiple mediation models by using Hayes’ PROCESS macro, Model 4 [79], setting bootstrapping to 10,000 resamples with 95% confidence intervals (CI). The independent variable was PA, the dependent variable was anxiety, and the mediators were SC and SE. We controlled for gender, education, and age. However, gender showed no significant difference, hence, it was excluded from covariates. The structural model showed structural paths (Figure 2). Bootstrap analyses are given below in Table 2, as are exhibited path coefficients for the parallel mediation model, indirect effects, 95% confidence intervals, and upper and lower limits of bootstraps.

The structural model (Figure 2) revealed that PA was positively associated with SE (b = 0.275, *p* < 0.01), which, in turn, was inversely related to anxiety of students (b = −0.172, *p* < 0.05), supporting the hypotheses that higher levels of PA increased the level of SE, and higher levels of SE led to reduced levels of anxiety. Independently, PA was inversely related to SC (b = −0.133, *p* < 0.05), which was positively associated with anxiety (b = 0.372, *p* < 0.001), supporting the predictions that higher levels of PA decreased level of SC, while lower levels of SC increased levels of anxiety. Furthermore, the indirect effect of SE (a1b1 = −0.0474) and the indirect effect of SC (a2b2 = 0.0498) indicate the significant mediation of SE and SC between PA and students’ anxiety.

Controlling for education and age, the association between PA and anxiety was significant (b = −0.176, *p* < 0.001). In support of the major study hypotheses that SC and SE would have mediating effects between PA and anxiety, a significant indirect association was established for both SE (b = −0.047; 95% CI = −0.0058, 0.0212) having a minimal effect size (0.0022), and SC (b = −0.049; 95% CI = −0.0099, 0.0235) having an effect size of 0.0024, which is also quite low. The proportion of the total effect of PA on anxiety that was predicted through SE was 34.5%. The proportion of the total effect of PA on anxiety that was predicted through SC was 18.2%. This means that 52.7% of the total effect was accounted for by these two mediators, while 47.3% of the relationship operated directly. Hence, it is supported that there is mediation in the relationship between PA and anxiety by SE and SC.

## 4. Discussion

Based on the data collected from Chinese students, this research validates a theoretical model of the connection between PA and anxiety. Findings provided evidence that the variables in the hypothesized model did predict anxiety, and the significant finding was that the effect of PA on anxiety not only operates directy but indirectly through SE and SC.

The study results contribute to the theory and literature by providing endorsement that inactive students were more likely to develop psychosocial problems, whereas active students showed higher levels of psychological health [22,23,24]. The results imply that mental health issues, such as anxiety, are influenced by PA. Findings of this study corroborate the arguments proposed by Rebar, Stanton [80] that PA is positively associated with physical, mental, and psychological outcomes, such as reductions in stress, depression, or anxiety. Endorsing our results, researchers have recently suggested that regular participation in sport and PA reduces stress, anxiety, and depression in university students [26,50,56,81]. The research indicates that, in terms of reducing anxiety, PA has the potential to assist university students in obtaining beneficial results.

Further, the multiple mediation model tested in the present study confirmed that SE mediated the association between PA and anxiety after controlling for age and education. Like many previous studies [50,56,82,83], we used PA as a predictor variable for developing mediation models in various contexts. Notwithstanding this, it was possible in this study to postulate SE as a mediator between PA and anxiety in a specific context. This is a crucial strength of the current study, as other studies have reported the correlational link between PA and other self-determined psychological variables or self-evaluation motives, such as self-efficacy [84] and self-esteem [82,85]. This result follows prior work indicating that students who engaged in more PA had high SE, leading to reduced anxiety [50,56]. The mediating role of SE implies that enhancement of the self has potential for the promotion of psychological health, if it is taught to university students. PA was found to influence SE positively. The finding was consistent with studies that publicized that physically active populations have satisfactory enhancement for their self [50,56] or their self-esteem [82], as compared to less active populations. On the flip side, anxiety is also an essential factor associated with SE and PA [86]. Our findings endorse past results which indicate that regular exercise has a positive influence on depression and high life satisfaction [87]. The results also supported the suggestion that SE mediated PA and anxiety. The mediation by SE of the association between PA and anxiety is consistent with the recent work of Kayani, Wang [56] in a South Asian country. It proves the importance of SE for university students in gaining positive self-views [29].

Multiple mediation analyses confirmed that SC also significantly mediated between PA and anxiety, after controlling for age and education. We found that students doing more PA were less engaged in SC, which is further associated with a reduction in anxiety. Past research in Western and Asian societies supports these results [50,56]. This finding is similar to research reported by Duarte, Stubbs [43], which described how an increased amount of PA lead to decreased SC. Comversely, there is a significant positive relationship between SC and anxiety associated with previous work [56]. In the present research, we observed that PA weakened the level of association between PA and SC [51,52]. Chen, Chen [86] confirmed that high SC leads to a high level of anxiety. Thus, students possessing less negative self-views have the potential to show less anxiety. Furthermore, PA impacts SC, which further diminishes anxiety, suggesting that PA depends on SC, to some extent, for transferring its effect to anxiety.

Findings from the present study have some implications for enhancing the self-system, especially in the context of China, where the self is considered as bi-cultural, containing the components of both individualistic and collectivistic cultures [59]. Pedagogical implications can also be drawn from the current study. This research shows a positive association between high PA and lower levels of anxiety, mediated by SE and SC. Thus, we recommend that teachers have to be educated with professional training in how to motivate students to participate more in PA in order to attenuate anxiety. Importantly, if teachers are trained to promote SE in their students, and to help students avoid SC, then the effects of PA on anxiety are likely to be enhanced. Moreover, the results of this research identified the significant role of PA in the reduction of anxiety and SC that often hamper students from achieving high academic performances. Indeed, anxious and self-critical students are likely to find it difficult to achieve academic results that reflect their true capability. Hence, we recommend that teachers learn to mitigate behaviors that are likely to increase PA and SC, thus reducing anxiety among students. Regarding the valuable benefits of PA, we recommend that the universities consider designing programs that can motivate students to adopt sport and PA. Overall, the findings also imply that interventions aimed at promoting PA and enhancing the self-system as a way to reduce anxiety would be a fruitful strategy.

## 5. Limitations and Future Research

Based on a thorough examination of the literature, to the best of our knowledge, the present study is the first to embrace key factors that lead to a reduction of anxiety in university students. Still, a few constraints exist in this study, which should be addressed in future work. First, given that China is a vast country with striking differences between various regions, the sample size is relatively small, and the use of a convenience sampling strategy could not produce a representative sample. Therefore, the generalizability of the findings from the current study is limited to a relatively small population in China. Further, the effect sizes for the mediating effect of SE and SC in this study were minimal, showing weak support for mediation of the variables. It may be due to the small sample size. Hence, developing the model with a large sample in future research may generate different results. Second, we conducted a cross-sectional study that does not show changes over time. Future researchers are advised to use longitudinal data as cross-sectional data is often criticized for common method bias. In addition, we have not checked each dimension of SE and SC which might be affected by low, moderate, or vigorous PA, and which may lead to different levels of anxiety. These limitations open a new approach for future researchers.

## 6. Conclusions

In the present study, we examined the relationship between PA and anxiety, with a particular focus on the intervening effect of SE and SC in a sample of Chinese university students. The results indicated that PA had a significant influence on SE and SC, which both significantly mediated the impact of PA on Chinese students’ anxiety. These findings emphasize the importance of PA for reducing anxiety and enhancing the self-system in Chinese culture.

## Figures and Tables

**Figure 1 ijerph-18-04468-f001:**
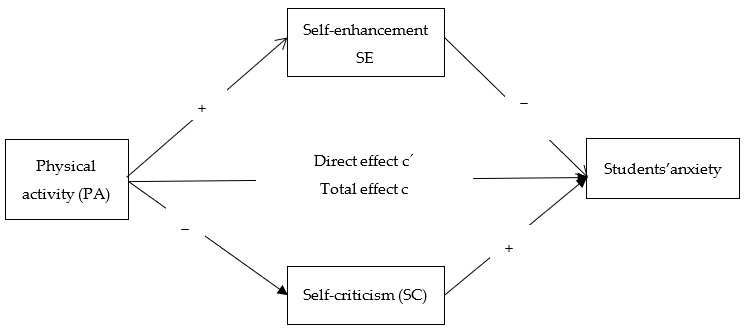
Study model.

**Figure 2 ijerph-18-04468-f002:**
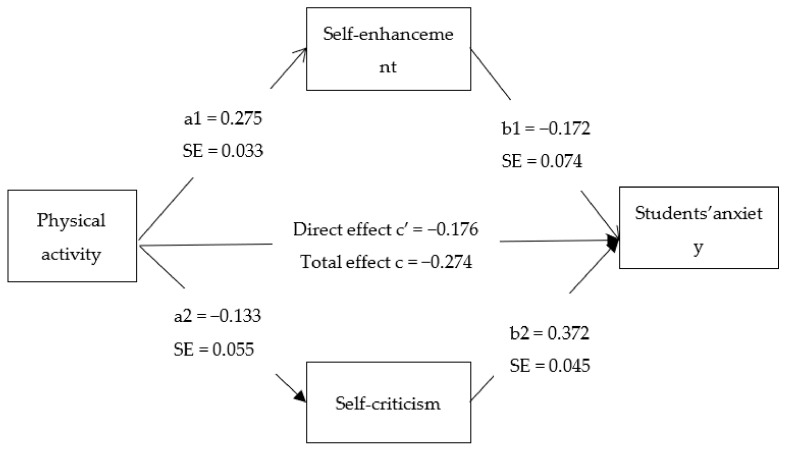
Structural model exhibiting mediation of SE and SC between PA and anxiety.

**Table 1 ijerph-18-04468-t001:** Correlations of study variables.

Variables	1	2	3	4	5	6
1. Education	-					
2. Age	0.087	-				
3. PA	−0.051	−0.218 **	-			
4. SE	0.103 *	−0.016	0.425 **	-		
5. SC	0.079	0.233 **	−0.137 *	0.011	-	
6. Anxiety	0.127 *	0.489 **	−0.309 **	−0.206 **	0.436 **	-

** Correlation is significant at the 0.01 level (2-tailed). * Correlation is significant at the 0.05 level (2-tailed). PA= physical activity; SE = self-enhancement; SC = Self-criticism.

**Table 2 ijerph-18-04468-t002:** Bootstrapping of mediation analyses**.**

Path	Beta	Boot-LLCI	Boot-ULCI	SE	T	*p*-Value
c = (a1b1 + a2b2 + c’)	−0.2740	−0.3693	−0.1788	0.0484	−5.6617	0.0000
c’	−0.1768	0.2731	−0.0805	0.0489	−3.6135	0.0004
IVM1 (a1)	0.2752	0.2090	0.3414	0.0336	8.1791	0.0000
IV-M2 (a2)	−0.1337	−0.2429	−0.0245	0.0555	−2.4085	0.0166
M1-DV (b1)	−0.1724	−0.3199	−0.0250	0.0749	−2.3017	0.0220
M2-DV (b2)	0.3722	0.2829	0.4616	0.0454	8.1980	0.0000
(a1b1 + a2b2)	−0.0972	−0.1643	−0.0410	0.0308	-	-
IV-M1-DV (a1b1)	−0.0474	−0.0918	−0.0058	0.0212	-	-
IV-M2-DV (a2b2)	−0.0498	−0.1018	−0.0099	0.0235	-	-

## Data Availability

Not applicable.

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
