# Peer review of "Physical Activity and Anxiety of Chinese University Students: Mediation of Self-System"

_ijerph, 2021, doi:10.3390/ijerph18094468_

Round 1
Reviewer 1 Report
This study investigated the mediating effect of self-system on the relationship between physical activity and academic anxiety in Chinese undergraduate students. The results showed the significant mediating effect of self-enhancement and self-criticism. This study contributes to our understanding toward the inter-relationships among physical activity, academic anxiety, and self-system. However, some concerns need to be addressed before this manuscript could be accepted for publication.
- When the authors used “academic anxiety”, the audience may think that this study has emphasized students’ anxiety regarding their academic performance. However, anxiety was measured in this study using the STAI Y-6 which was not specifically used to measure academic anxiety. Therefore, the authors should consider to change the title of this study; otherwise, it may mislead the audience.
- The authors interpreted the pathway from PA, through self-system, to anxiety as the partial mediation. However, when the complete and/or partial mediation is conventionally used by Baron and Kenny to describe the indirect effect of the independent variable on the dependent variable through the mediator(s), Hayes has challenged this concept. Particularly, when the PROCESS was used to test the mediation in this study, I assumed that the authors agreed with Hayes, and that the idea of the complete/partial mediation obviously conflicts Hayes’. Therefore, I recommend that the authors should seriously think whether the partial mediation would be used to interpret the results in this study.
Andrew F. Hayes (2009) Beyond Baron and Kenny: Statistical Mediation Analysis in the New Millennium, Communication Monographs, 76:4, 408-420, DOI: 10.1080/03637750903310360
- I feel that the Results section needs to be re-structured. I would suggest to describe the demographic information the participants (please move part of 2.1 Participants to this section), then correlation matrix, followed by the model fit of the overall model and the results of the tested pathways. By doing so, the audience should be able to follow the results more easily. In addition, Table 2 is unclear to me. It may be not helpful to understand the results of the mediation. I would suggest the authors to report the results based on the PROCESS output or Hayes’ book.
- When the PROCESS was used to test the mediation, the authors set the number of bootstrap samples at 2,000. However, Hayes and Scharkow (2013) have suggested 10,000 would be more appropriate. A reference should be provided to support the use of 2,000.
Hayes, A.F., & Scharkow, M. (2013). The relative trustworthiness of inferential tests of the indirect effect in statistical mediation analysis: Does method really matter? Psychological Science, 24, 1918–1927. doi:10.1177/0956797613480187
- The authors repeatedly mentioned “bi-cultural”; however, the impact of this cultural context was not discussed on Chinese students’ mental health. Did the authors believe that having individualistic and collectivistic cultures simultaneously would affect the development of self in Chinese students? More interpretation would be needed.
There are a few minor comments listed below.
- When the abbreviations of PA and AA were defined, they were not used throughout the manuscript. The authors should carefully go through the manuscript and replace physical activity and academic anxiety.
- Some abbreviations were not defined, such as EFA, CFA, etc.
- In 2.3 Data Analysis, the authors describe how to test the mediation using the PROCESS, but did not define M, X, and Y. This may confuse the audience if they are not familiar with the PROCESS.
- In 2.2.1 Physical activity, the scoring was not clear. For example, how was the scaled score calculated? When Cho (2016) did not provide the cutoff point for PA levels, how were these cutoffs developed? In addition, the reference seems to be inappropriately cited as the authors cited Cho 2014 and it’s Cho 2016 in the References.
- The citation format was not consistent. In addition, some references may be listed with some missing information, such as Ref 8, 16, 17, 19, 24, 60, 70, 78….
Reviewer 2 Report
Of the 89 citations, six were more than 20 years old, one (#42) The psychoanalytic study of the child, was 47 years old!!!!! Surely the authors could find more up-to-date studies to use. Two others were over 32 years old and still two others were over 22 years old. Researchers found individuals doing more physical activity have greater emotional stability and are less prone to depression. This has long been the case. The authors state over and over again this is particularly true in Chinese students. Further studies show positive change in moods, self concept, including self-efficacy and enhancement of mood "in the Chinese population." Authors suggest teachers need to be educated with professional training on how to motivate students to be more physically active to mitigate behaviors leading to academic anxiety. Studies imply that implementations aimed at promoting physical activity to reduce academic anxiety would be a fruitful strategy - stating the Chinese government should consider designing these programs. All this to emphasize Chinese students are "different" and this research affects only Chinese students. I beg to differ - all university students face basically the same obstacles - need more physical activity to fend off academic anxiety and feelings of depression. The information is good but it is not necessary to repeat it over and over again, especially using such dated research in some case
Reviewer 3 Report
I appreciate the opportunity to review this interesting manuscript. It is a non-experimental work with Chinese university students. It is intended to know to what extent self-enhancement and self-criticism mediate the effect of physical activity on academic anxiety. The responses of 305 students were treated using the PROCESS software, statistically controlling the effect of variables such as age or education. The results confirmed all the hypotheses raised. I think the manuscript addresses a topic that is of interest to IJERPH readers. The research objective is consistently justified and the manuscript well organized and developed, and well written. However, there are some aspects that could be considered by the authors with a view to a potential improvement of the current form of the manuscript. Most of the suggestions are minor changes, but important to address.
Abstract. I think the authors might consider including the effect of the two mediating variables on the dependent variable as well (this part of the model is omitted).
Theorical Background. This first part of the manuscript is very well organized and very well written. However, I think that some space should be dedicated to present the theoretical model used in the consideration of the two mediating variables (self-enhancement and self-criticism), since all that is done is to report the possible links with the independent variable and with the clerk. The theoretical model provides a necessary meaning to these two constructs and based on it the results should be discussed. On the other hand, the authors state the hypotheses of the model (if there is a statistically significant relationship and its sign), but they do not state the magnitude of the effects (they should). Finally, it seems that a partial mediation model is initially proposed for both mediating variables, why should a total mediation model not be hypothesized (that is, that the effect of PA on AA is only indirect). Some justification should be included here.
Method.
Participants. The authors of the manuscript correctly point out in the limitations section the possible non-representativeness of the sample and the difficulty in generalizing the results. Well, this is hopeless at this point. What the authors could do is provide information on other characteristics of the sample that could help the reader to compare these data with other data derived from other studies carried out with students from China or elsewhere. I think figure 2 is superfluous.
Data Analysis. It is not clear how possible missing values within the final sample were handled. On the other hand, it is not indicated how the sizes of the effects found will be valued (in fact, it is not done). Also, I do not quite understand why when considering several mediating variables it is not done through a SEM model or a path Analysis instead of using PROCESS. I suggest including some justification. Finally, the authors of the manuscript indicate criteria to assess the fit indices that, in my opinion, are lax. Specifically, SRMR and RMSEA should be less than or equal to 0.05; CFI and TLI should be greater than or equal to 0.95.
Results. No information is provided on effect sizes. To refer to them, if they consider it appropriate, the authors of the work can consult:
Funder, D. C., & Ozer, D. J. (2019). Evaluating effect size in psychological research: Sense and nonsense. Advances in Methods and Practices in Psychological Science, 2, 156–168. doi:10.1177/2515245919847202
Discussion. This section is well elaborated. However, it could be improved if the effects are discussed along with the significance of their sizes. For example, the indirect effect of PA on AA, through SE, is very small (.047) and its size is minimal (.0022). Therefore, although there is an indirect effect (partial mediation through SE), can it be said that SE is really a sufficiently mediating variable to support the statements made in the discussion and in the conclusions? And like this, the rest of the effects (direct and indirect). Another question that could enrich the manuscript would be to explore which variables not included in the model would mediate the direct effect that was found. Finally, it would be important to provide the explained variance of AA, and which corresponds to the mediation (of each variable and jointly). This is important to understand the importance of mediating variables.
This is all. I congratulate the authors as this is a good and very interesting manuscript. If it is possible to attend to some of the suggestions, the better.
Round 2
Reviewer 1 Report
The paper has been significantly improved. I only have few minor comments now.
1. When the authors used “wellbeing” throughout the manuscript, I would suggest to revise “well-being” on the first line of Page 4 to be consistent.
2. I previously suggested to report the results based on the PROCESS output; however, I didn’t mean that the complete output table should be completely pasted. I apologized for this misunderstanding if I was not clear enough. As it may be not appropriate to copy and paste the whole output table of the PROCESS analysis (Page 8-19), I recommend the authors to summarize the results. Specifically, as the mediating effect is the pathway of interest in this manuscript, I recommend that the authors could create a table to report the indirect effect. In addition, a brief description regarding the results should be provided in the main context.
3. 5. Limitations and Future Research: Line 7, Se and Sc should be SE and SC.
4. Figure 2: When I read the output file of the PROCESS, a1 and c should be -.134 and -.177, respectively.

Author Response
Reply to Review-1 (ROUND-2)
Respected Reviewer, thank you so much for providing us such an effective feedback on the article entitled “Physical Activity and Academic Anxiety of Chinese University Students: mediation of self-system”. A reply to the review report is given below.
Comments and Suggestions for Authors
The paper has been significantly improved. I only have few minor comments now.
- When the authors used “wellbeing” throughout the manuscript, I would suggest revising “well-being” on the first line of Page 4 to be consistent.
Reply: Respected reviewer, your comment is highly appreciated. The change has been made according to your valuable suggestion. These can be tracked in the manuscript.
- I previously suggested reporting the results based on the PROCESS output; however, I didn’t mean that the complete output table should be completely pasted. I apologized for this misunderstanding if I was not clear enough. As it may be not appropriate to copy and paste the whole output table of the PROCESS analysis (Page 8-19), I recommend the authors to summarize the results. Specifically, as the mediating effect is the pathway of interest in this manuscript, I recommend that the authors could create a table to report the indirect effect. In addition, a brief description regarding the results should be provided in the main context.
- 5. Limitations and Future Research: Line 7, Se and Sc should be SE and SC.
Reply: Respected reviewer, your comment is highly appreciated. The change has been done.
- Figure 2: When I read the output file of the PROCESS, a1 and c should be -.134 and -.177, respectively.